# Metastatic Lobular Breast Cancer Mimicking Colitis

**Renata Reis Figueiredo** [1,*], **Tatiana Strava Correa** [1], **Carlos Henrique dos Anjos** [1]
**and Heinrich Bender Kohnert Seidler** [2]

[1]  Clinical Oncology Department, Sírio Libanês Hospital, Brasilia 70200-730, Brazil;
    tatiana.scorrea@hsl.org.br (T.S.C.); carlos.hanjos@hsl.org.br (C.H.d.A.)
[2]  Pathology Department, Brasiliense Lab, Brasilia 70390-125, Brazil; hseidler@gmail.com
*  Correspondence: renatareisfigueiredo@gmail.com; Tel.: +55-61-99922-8005

**Abstract:** Breast cancer is the most frequent cancer diagnosed in women in the world regardless of race or ethnicity. About 10% of invasive breast carcinomas are lobular subtype. The loss of the E-caderin expression that occurs in lobular carcinoma leads to a higher risk of metastases in membranes (meningeal, pleural, peritoneum) and gastrointestinal and/or endobronchial mucous, which may lead to several odd symptomatology. We report a 79 years old female patient with lobular breast cancer associated to CDH1 germline mutation. She was diagnosed with breast cancer in December 2016 after noticing a right-armpit nodule whose pathological examination demonstrated an immunohistochemistry profile compatible with lobular breast carcinoma metastasis and had estrogen receptors 98%, progesterone receptors < 1%, ki67 25%, negative her2 score. Family history of only one paternal uncle with stomach cancer. After two lines of hormone therapy, she had disease progression and started oral chemotherapy with capecitabine. In a few weeks, the patient had refractory diarrhea. At the beginning, it was defined like colitis chemotherapy related. However, the clinical features showed necessity of further investigation. Then, she was diagnosed with CDH1 germline mutation after massive progression at gastrointestinal mucous. This case made possible to inform the family about risk of germline mutation and necessity of genetic counseling.

**Keywords:** lobular carcinoma; colitis; CDH1 mutation

## 1. Introduction

Breast cancer is the most frequent cancer diagnosed in women in the world regardless of race or ethnicity. In addition, it is the leading cause of cancer-related death in women of Hispanic origin and the second cause of cancer-related death in white, black and Asian [1]. Even though the two most common breast carcinomas arise from terminal ductal lobular units, the frequency of ductal and lobular carcinomas are quite different. Approximately 70% to 80% of invasive breast carcinomas are ductal and 5% to 10% are lobular [2]. Although they are histologically different diseases, both can generate metastases for liver and bones, but lobular carcinomas are more frequently involved in gastrointestinal metastases, peritoneum and gynecological organ [3]. Understanding the differences between these diseases is extremely important for defining strategies, treatment and monitoring with special attention to their particularities. We reported a case of lobular breast cancer with extensive metastasis in the gastrointestinal tract.

## 2. Case Presentation Section

A 79-year-old female patient was diagnosed with breast cancer in December 2016 after noticing a right-armpit nodule whose pathological examination demonstrated an immunohistochemistry profile compatible with lobular breast carcinoma metastasis and had estrogen receptors 98%, progesterone

receptors < 1%, ki67 25%, negative her2 score. Family history of only one paternal uncle with stomach cancer. She had metastatic disease *de novo* with lesions in lymph nodes and dermis. The patient started the first line of treatment with fulvestrant associated with denosumab until September 2017. At that time, she evolved with progression of lung disease and started second line of treatment with exemestan and everolimus. She remained with this treatment for 9 months when presented a new progression of lung disease, lymph nodes and dermis. In June 2018, she started oral chemotherapy with capecitabine. In a few weeks, the patient complained of grade 2 diarrhea refractory to loperamide. Initially, the symptom was associated with toxicity related to treatment. However, the patient maintained refractory diarrhea and evolved with vomiting, in addition to the need for intravenous supportive treatment. She received parenteral nutrition for 3 weeks due to caloric protein malnutrition secondary to diarrhea. Colonoscopy demonstrated colon mucosa with numerous ulcers (Figure 1). Serial biopsies were performed showing neoplastic cell infiltration in colonic mucosa. Anatomopathological study was compatible with breast cancer infiltration and revealed poorly differentiated carcinoma with solid cell morphological pattern slightly cohesive with signet ring cells (Figure 2). Immunohistochemistry demonstrated positive AE1/AE3, focal BRST-2/GCDFP-15 positive, negative CK-20, positive CK-7, positive mammaglobin and positive estrogen receptor (Figure 3).

The patient started intravenous chemotherapy with cyclophosphamide and liposomal doxorubicin and evolved with resolution of gastrointestinal complaints. She underwent 4 cycles of treatment and then had a new progression of peritoneum disease (Figure 4). She received new chemotherapy with eribulin from November 2018 to July 2019, suspended due to progression of liver disease and lymph nodes. She also received palbociclib associated with fulvestrant for two months, but due to further disease progression, she restarted chemotherapy with weekly paclitaxel in September 2019. Somatic genomic panel of the tumor (FOUNDATIONOne Liquid) was performed and demonstrated the presence of CDH1 mutation (MAF = 30%). After genetic counseling, the germ test (Invitae) was also performed and a germline mutation was found in CDH1. Then, the patient's daughters were instructed to perform the mutation testing.

The patient evolved with disease progression, clinical deterioration, being established exclusive palliative care and died in October 2019.

Patient consent was obtained for this case report.

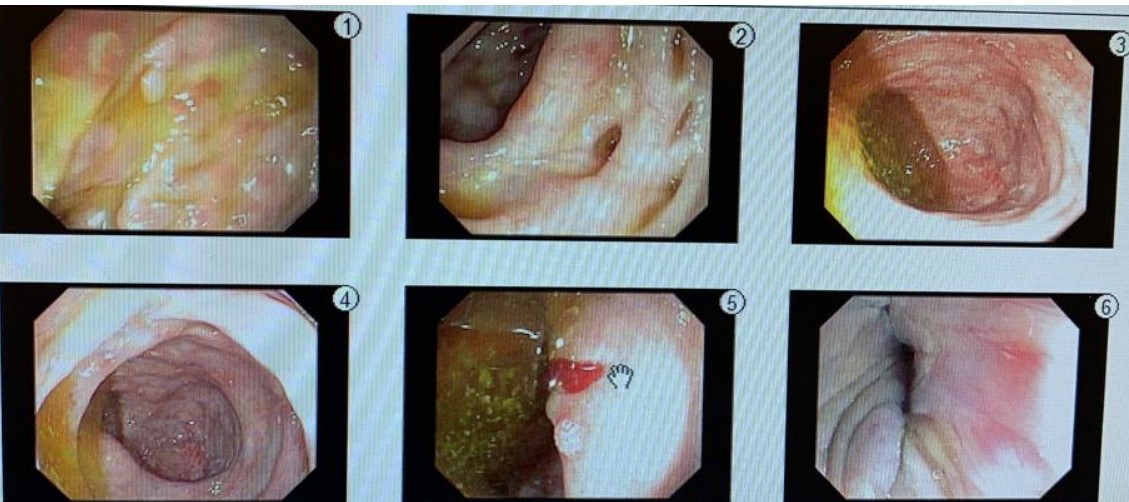

**Figure 1.** Colonoscopy images showing diffuse mucosa erosions.

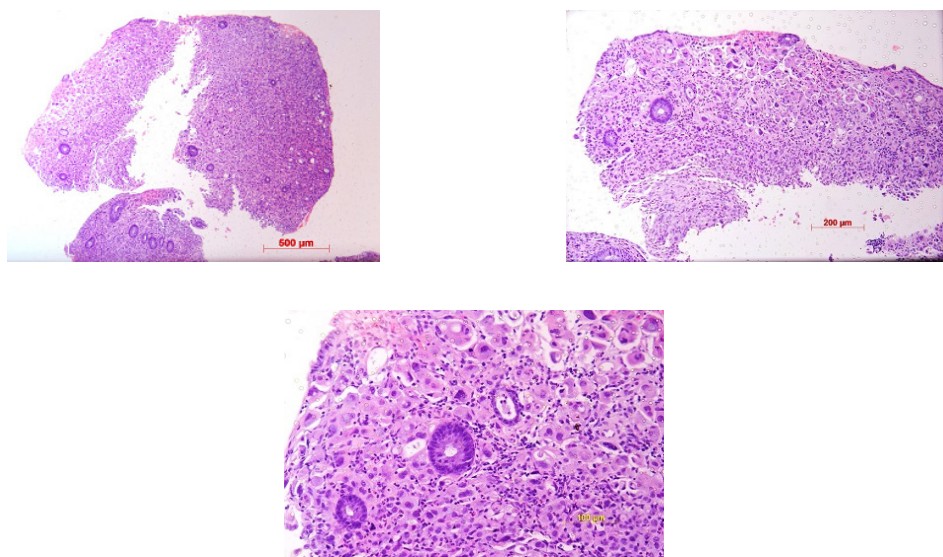

**Figure 2.** Histological evaluation reveals poorly differentiated malignancy invading colonic and rectal mucosa.

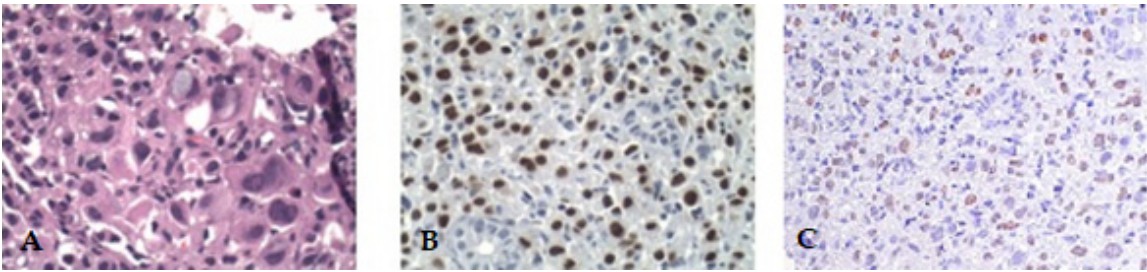

**Figure 3.** Immunohistochemistry of the colon biopsy showing a result compatible with breast cancer infiltration. (**A**): Hematoxylin-eosin staining, (**B**): GATA3 expression (L50–823 clone), (**C**): Estrogen receptor expression (EP1 clone).

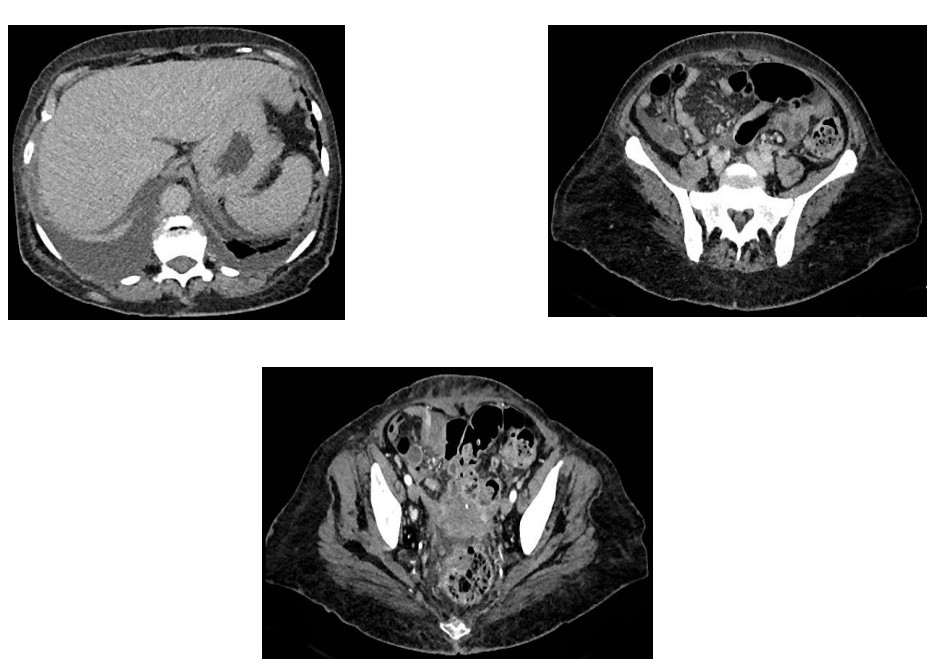

**Figure 4.** CT images showing peritoneum disease progression.

## 3. Discussion

Studies conducted in the United States indicated a 65% increase in the incidence of invasive lobular carcinoma from 1987 to 1999 while, in the same period, the incidence of invasive ductal carcinoma increased only 3% [4]. Hormone replacement therapy was pointed out as probable cause of this increased incidence [5]. After 1999, age-adjusted incidence rates decreased in both groups [4].

It is known that patients with invasive lobular carcinomas are diagnosed 3 years above those with invasive ductal carcinoma, with median age of 57 years at diagnosis [5]. In addition, lobular carcinomas are usually diagnosed in more advanced states, with larger tumors and more often associated with lymph node involvement [4].

Invasive lobular carcinoma is more strongly associated with early menarche, late menopause and older age at the birth of the first child. More than 90% of invasive lobular carcinomas are positive for estrogen receptors and are randomly classified as luminal A. The her2 overexpression is rare, present in only 3% to 5% of classic invasive lobular carcinomas but may be present in up to 80% of cases in the most aggressive pleomorphic subgroup [6].

Regarding genetic risks, four genes of high penetrance are tested in clinical practice when genetic susceptibility to breast cancer is suspected: BRCA1, BRCA2, TP53 and CDH1. Germ mutations in BCRA1 and TP53 are predominantly associated with invasive ductal carcinoma, while BCRA2 mutations are associated with both ductal and lobular tumors. Mutations of the CDH1 gene, which encodes E-caderin adhesion protein, are particularly interesting because they are associated with invasive lobular carcinoma but never to ductal carcinoma [4]. In the case presented, the patient has germ CDH1 mutation.

Patients with lobular carcinoma have higher risk of metastases for gastrointestinal mucosa, peritoneum, pleura, meninges when compared to metastatic ductal carcinoma. This is a consequence of the loss of the E-caderin expression [7], which is considered the most consistent feature reported to invasive lobular carcinoma, demonstrated in up to 90% of cases. E-caderin is a calcium-dependent transmembrane protein involved in the adhesion of epithelial cells and its absence predisposes to neoplastic proliferation. This deregulation results from somatic mutations in the CDH1 gene of chromosome 16q22.1 reported in 30% to 80% of invasive lobular carcinomas, as well as loss of heterozygosis of CDH1. However, E-caderin positivity alone does not exclude lobular neoplasia, since not all lobular carcinomas carry CDH1 mutations [6].

The diarrhea complain is very common in oncologic patients. There are multiple causes, including intestinal resection, infections, radiation and systemic anticancer treatments [8]. The signs and symptoms of metastases for gastrointestinal mucosa are also variable. The main ones are abdominal pain, nausea, vomiting, bleeding and weight loss. Unabsorptive diarrhea, as seen in this case, is not a common symptom, which drew attention to it, initially confounding the correct diagnosis of disease progression with diarrhea secondary to infection and drug toxicity. Colonoscopy was important for diagnosis, confirmed by pathology.

Heredity is uncommon but can be found as a secondary tumor in families with hereditary diffuse gastric cancer syndrome [6], as in the case of this patient who, despite the family history not very relevant, presented the germ CDH1 mutation.

## 4. Conclusions

Knowledge of the clinical evolution of lobular breast cancer and its particularities are important for distinction between toxicity related to treatment and refractoriness of the disease. The loss of the expression E-caderin that occurs in lobular carcinoma leads to a higher risk of metastases in membranes and mucous membranes that can lead to several symptomatology that oncologists should be attentive to diagnose. In addition, this case demonstrates the importance of germline genetic evaluation, in order to suggest genetic family counseling.

**Author Contributions:** The case report was written by all authors. All authors have read and agreed to the published version of the manuscript.

**Funding:** This research received no external funding.

**Conflicts of Interest:** The authors declare no conflict of interest.

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
