# Peer review of "Metastatic Lobular Breast Cancer Mimicking Colitis"

_reports, doi:10.3390/reports3030020_

Round 1

Reviewer 1 Report

The presented case of disseminated lobular carcinoma is based on rare colonic involvement. The case needs to be improved to have a teaching potential. The figures to show both primary carcinoma and its comparsion with metastases. The summary of causes of diarrhoea in cancer patients. The summary of different places of metastases from lobular carcinoma.

Author Response

To Reviewer 1

The case was restructured according to the template's rules.

English writing has been improved.

The conclusion reinforced the importance of publishing the case.

The main causes of diarrhea in cancer patients have been described.

Thank you

Reviewer 2 Report

This is an interesting case report study. However, the paper is not well written and structured. In my opinion, the manuscript has some shortcomings in regards to text. Below I have provided some comments on the text as it is often vague and not well written.

Key critical points are:

a) Abstract. P1 and L8-9: the word "membranes" is repeated often. The authors should write better.

b) Background. P1 and L17-L18: Frequency percentages are exact, however both ductal and lobular carcinomas arise from terminal ductal lobular units (TDLUs). This is a fairly frequent and serious mistake. Please correct.

c) Case Presentation. The figures have no legend. Colonoscopy images (figure 1) are approximate and not accurate. Histology images (figure 2) are devoid of scale bars and magnification.

Given these shortcomings, the manuscript requires major revisions.

Author Response

To Reviewer 2

The case was restructured according to the template's rules.

English writing has been improved.

The error about the origin of breast carcinomas has been fixed.

Figure 1 shows colonoscopy images. We know that the image does not have a good resolution but the exam was performed in another service and we did not have access to better quality images. We are sorry.

Figure 2 is reported exactly as the original exam and does not have a scale bar. We can ask the pathology lab about it. 

Thank you

Round 2

Reviewer 1 Report

The work was improved. 

Author Response

We improve the pathology information as requested.

Reviewer 2 Report

The authors have addressed most of the points raised.

Just a point:

- Figure 2. Add the scale bar and a detailed caption to the figure. For example (A)..., (B)... and (C)... 

This is essential and meaningful.

Author Response

(The authors gave the same response as above.)
